# Understanding the Effects of Post-Deposition Sequential Annealing on the Physical and Chemical Properties of Cu$_2$ZnSnSe$_4$ Thin Films

**Diana-Stefania Catana [1], Mohamed Yassine Zaki [2] [iD], Iosif-Daniel Simandan [2], Angel-Theodor Buruiana [2] [iD], Florinel Sava [2] and Alin Velea [2],* [iD]**

[1] Faculty of Physics, University of Bucharest, Atomistilor 405, 077125 Magurele, Romania; diana.c1098@gmail.com

[2] National Institute of Materials Physics, Atomistilor 405A, 077125 Magurele, Romania; yassine.zaki@infim.ro (M.Y.Z.); simandan@infim.ro (I.-D.S.); angel.buruiana@infim.ro (A.-T.B.); fsava@infim.ro (F.S.)

[*] Correspondence: alin.velea@infim.ro

**Abstract:** Cu$_2$ZnSnSe$_4$ thin films have been synthesized by employing two magnetron-sputtering depositions, interlaced with two sequential post-deposition heat treatments in low vacuum, Sn+Se and Se–rich atmospheres at 550 °C. By employing successive structural analysis methods, namely Grazing Incidence X–Ray Diffraction (GIXRD) and Raman Spectroscopy, secondary phases such as ZnSe coexisting with the main kesterite phase have been identified. SEM peered into the surface morphology of the samples, detecting structural defects and grain profiles, while EDS experiments showed off–stoichiometric elemental composition. The optical bandgaps in our samples were calculated by a widely used extrapolation method from recorded transmission spectra, holding values from 1.42 to 2.01 eV. Understanding the processes behind the appearance of secondary phases and occurring structural defects accompanied by finding ways to mitigate their impact on the solar cells' properties is the prime goal of the research beforehand.

**Keywords:** CZTSe; thin films; chalcogenides; secondary phases; absorber layer

## 1. Introduction

Chalcogenide thin film solar cells (TFSCs) are considered one of the first–hand candidates in the search for the up–and–coming photovoltaics industry. Sustainable and eco–friendly production practices are considered one of the main assets of this emerging sector. Therefore, creating an environmentally friendly technology of solar cells with earth–abundant constituents and low manufacturing costs, while achieving high conversion efficiencies, has become an even more feasible goal due to the emergence of chalcogenide thin films. Being part of a group of earth–abundant and non–toxic materials, chalcogenides also exhibit a high versatility in further solar cell–based preparation techniques [1]. Cu$_2$ZnSn(S,Se)$_4$ (CZTS, CZTSe, CZTSSe) displays remarkable optoelectronic properties such as a tunable bandgap [2] and a high absorption coefficient [3]. As a p–type semiconductor with a direct optical bandgap ranging between 0.9 eV and 1.1 eV and an absorption coefficient estimated to an order of $10^4$ cm$^{-1}$ [2,3], CZTSe films demonstrate optimal characteristics for possible TFSC implementations [4]. In the last years, it has also been probed for radiation hardness tests, thus making the list of promising space–operable materials [5].

Despite its acclaimed features, the structural complexity of the CZTSe compound requires a thorough assessment. A shortcoming that widens the bandgap and consequently lowers the efficiency of the cell consists of detrimental effects represented by secondary phases and lattice defects that can act as trap states [6,7]. CZTSe can be found in three tetragonal structures, respectively kesterite, stannite and pre–mixed Cu–Au (PMCA), where

kesterite showed an increase in PCE in the last years [8,9]. Due to a narrow thermodynamic stability span [10] and the inevitable presence of secondary phases, which are the most common hurdles encountered during synthesis, achieving a high–yielding J–V curve can become a real challenge [11]. The highest reported PCE of a CZTSe cell is roughly around 12.6% [12], which is lower than older, less sustainable and moisture–stable technologies such as CIGS, organic films and the newer inorganic and organic perovskites. These films are topping efficiencies close to 17–20% (for CIGS) and 22–24%, with the perovskites being reported and expected to exceed the Shockley–Queisser limit [13,14]. The improvement of the power conversion efficiency in CZTSe has caught the attention of the materials physics community and it has been undergoing intense research.

A long series of physical and chemical methods, carried out in vacuum or non–vacuum conditions, has been put to use to prepare kesterite thin films. Here we mention spin coating [15], aqueous synthesis [16], magnetron sputtering [17], thermal evaporation [18] pulsed laser deposition (PLD) [19], solvothermal synthesis [20] and mechanosynthesis [21]. Magnetron sputtering is considered one of the better vacuum deposition procedures because it provides a homogeneous coating, high–caliber adhesion to the substrate and enhanced maneuverability of in–situ physical parameters [22]. If prepared using a route that includes stacked layers [23], in order to ensure the reaction between the layers, after every deposition required to achieve the CZTSe material, post–deposition annealing within a 300–550 °C range is recommended [23]. The volatility factor of mainly Sn, and rarely Se and Zn compounds, imposes a huge drawback as well. This way, the usually off–stoichiometric concentrations can be mended and the formation of secondary phases better controlled. $ZnSe$, $CuSe$, $Cu_2Se$, $SnSe$ and the ternary $Cu_2SnSe_3$ (CTSe) are among the most detected secondary phases that co–exist with the main phase [24,25] and the efforts to reduce their unwanted existence is a priority in the field of kesterite TFSC production.

The current study begins with a meticulous description of a complex thin film growth procedure. Firstly, four stack configurations were deposited ($Cu/Sn$, $Cu_2Se/SnSe_2$, $Cu/SnSe_2$, $Cu_2Se/Sn$) on two different substrates. Next, after undergoing a Sn+Se atmosphere heat treatment, the samples were inserted in the magnetron chamber for a second deposition involving a ZnSe layer sputtered on top of the previous structure. Finally, a selenization was performed, culminating with the full preparation of eight CZTSe films. Detailed investigations on morphological, structural and optical properties of the obtained samples were performed.

## 2. Materials and Methods

Eight samples were prepared on two different substrates, namely Mo/SLG and SLG, each undergoing two sets of depositions and post–deposition heat treatments. There are four target configurations sputtered in order to obtain the first layers, as shown in Figure 1a. Hence, we labeled the samples two by two as Z1, Z2, Z3, Z4, corresponding to $Cu/Sn$, $Cu_2Se/SnSe_2$, $Cu/SnSe_2$, $Cu_2Se/Sn$, respectively, summing up to a total of eight different samples. Afterwards, each sample underwent a first annealing in an Sn and Se atmosphere. During the second deposition, a 150 nm ZnSe layer was stacked on top of previous structures. The synthesis operation was completed with a final post–deposition selenization. Figure 1b exhibits the four–step synthesis approach (two depositions intertwined by two complementary post–deposition heat treatments).

A Gencoa 3G Circular Magnetron array coupled to T&C Power Conversion AG 0313 RF (Radio Frequency) and AJA DC sources was used for the magnetron sputtering depositions. While during all depositions, the Ar flow was kept at a constant 30 standard cubic centimeters per minute (SCCM) volumetric rate and the pressure inside the magnetron chamber was maintained at $5 \times 10^{-3}$ bar, the power applied on the magnetron guns varies for each of the four configurations. Table 1 presents the parameters of each deposition (gun power, layer thickness and deposition rate).

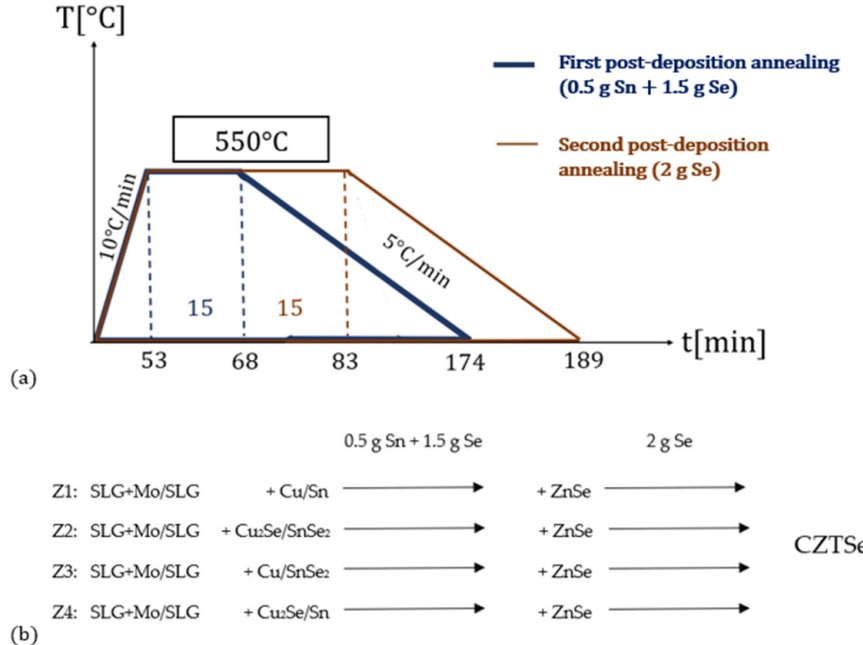

**Figure 1.** Synthesis steps. (**a**) Post–deposition heat treatments and (**b**) schematic summary of the entire CZTSe synthesis process.

**Table 1.** Parameters during the first magnetron–sputtering deposition.

| Parameters | Z1 | | Z2 | | Z3 | | Z4 | |
|---|---|---|---|---|---|---|---|---|
| | Cu | Sn | $Cu_2Se$ | $SnSe_2$ | Cu | $SnSe_2$ | $Cu_2Se$ | Sn |
| Gun power (W) | 24 | 14 | 16 | 16 | 14 | 16 | 17 | 15 |
| Thickness (nm) | 100 | 200 | 100 | 200 | 100 | 200 | 100 | 200 |
| Deposition rate (Å/s) | 0.5 | 0.5 | 0.5 | 0.26 | 0.5 | 0.26 | 0.5 | 0.26 |

The temperature inside the magnetron chamber was held to room temperature values, between 20 °C and 25 °C. We opted for a constant substrate holder rotation to ensure uniformity, homogeneity and isotropy of the elemental spread.

After this deposition, the samples are ready for the first post–deposition heat treatment. A GSL 1600X tubular furnace was used to enable low vacuum conditions ($10^{-2}$ Torr) with a steady argon flow (10 SCCM). During such processes, boundary reactions occur between the compound layers, thus enhancing their chemical stability. The samples were all annealed in Sn– and Se–rich atmospheres with different concentration ratios, 0.5 g of Sn and 1.5 g of Se. The temperature inside the furnace follows a trapezoidal curve, rising from room temperature with a constant growth rate of 10 °C/min until it reaches a steady plateau of 550 °C for 15 min. With a shallower cooling rate of 5 °C/min that brings the temperature back to ambient values, the entire heat treatment amounts to 174 min (Figure 1a).

The second–deposited layer of ZnSe has an estimated thickness of 150 nm for the entire set of 8 samples. We employed the same circular magnetrons along with an RF source and applied a gun power of 15 W. For a better uniformity of the elemental spread, the substrate holder was also rotated with a constant angular frequency. The thermodynamic parameters (pressure and temperature) inside the chamber were reported to have maintained the same previous values. Ultimately, the total thickness of the stack became 450 nm.

Afterwards, a final post–deposition selenization heat treatment with 2 g of Se powder was performed. The steady plateau of 550 °C reached along the temperature trapezoidal curve lasts for 30 min, thus the process spanning over 189 min (see Figure 1a).

As we completed the synthesis of the samples, compositional and morphological analysis techniques were employed for this study.

A Zeiss EVO 50 XVP scanning electron microscope (SEM) was used to map the surface morphology of each film. The coupled energy dispersive spectroscopy (EDS) device, an AXS Microanalysis GmbH model from Bruker, conducted the identification of the elemental concentration in all samples. Next, Raman spectroscopy was employed to attain precise phase purity and secondary phase identification. The Raman spectra were recorded with a He–Ne laser–based Jobin–Yvon spectrometer using an excitation wavelength of 633 nm, annexed to a confocal Olympus $100\times$ microscope.

The crystalline structure of the CZTSe films and, subsequently the average crystallite size, was investigated by the Grazing Incidence X–ray Diffraction (GIXRD) technique with the help of a Rigaku SmartLab X–ray diffractometer. The wavelength of the CuK$\alpha$ radiation was 1.5406 Å, being emitted under $0.5°$, while the bouncing detector was set to move in a Bragg–Brentano geometry. Moreover, we also built the crystal structure (schematic mesh of atoms and bonds) using the pre–established VESTA color chart: blue dots for Cu(2+), gray for Zn(2+), purple–gray for Sn (2−) and green for Se(2−).

The latest measurements involve the calculation of the optical bandgaps of the four SLG–deposited films by the Tauc Plot method [26]. In order to obtain these results, the transmission spectra were collected with a V–VASE Woollam Spectroscopic Ellipsometer equipped with a high–pressure Xenon discharge lamp placed inside an HS–190 monochromator covering a wavelength range between 350 nm and 1700 nm.

## 3. Results

### 3.1. GIXRD Analysis and the Determination of the Average Crystallite Size

The GIXRD diffractograms, plotted for all samples, can be observed in Figure 2. Several peaks can be distinguished, with the main highest peak found at $2\theta$ values between $27.24°$ and $27.32°$, with an average of $27.28°$ encountered in four of the eight samples. This corresponds to the (1 1 2) orientation for the dominant $Cu_2ZnSnSe_4$ kesterite phase, followed by (2 0 4) at $44.72°$ and (3 1 2) at $53.01°$ [27]. Thus, by aligning the stacked diffractograms with ICDD markers, we identified the afore–mentioned CZTSe kesterite phase (ICDD card 04–019–1866) but also the characteristic peaks of ZnSe phase (ICDD 01–071–5977). In Figure 2, we have included the ICDD histograms for the identified phases. The crystal structures of kesterite and ZnSe can be described in the middle–up and middle–bottom sections of the figure, respectively.

The main CZTSe peaks are accompanied to the left by a thin hump aligned between $26.98°$ and $27.03°$ that might occur as a cause of secondary phases such as $Cu_2SnSe_3$ or even $Cu_2Se$. The lack of Cu–rich binary and ternary phases, that according to Yao et al. [28] should be present in the immediate vicinity of the main CZTSe phase, implies a poor reactivity between the Cu, Se and Sn precursors and accentuated volatility. Ternary phases such as CTSe, and binary phases like $Cu_2Se$, CuSe are therefore usually quoted to overlap with the double Bragg angle marking kesterite. As another gauging of well–known assumptions, neither SnSe nor $SnSe_2$ are present in the examined diffractograms. The theory assigns the small peak at $30.46°$ and $31.32°$ to a superposition of coexisting secondary phases, such as ZnSe, CTSe, CuSe and SnSe [27,29], but, in our case, the peak might just be CTSe. Sn–based phases are rare by default and briefly visible, a consequence of the accentuated volatility of Sn at high temperatures, as those involved during the post–deposition annealing procedures at 550 °C.

ZnSe can be identified by the peaks at $53.64°$, $45.26°$–$45.32°$ and $71.8°$–$72.3$ as a protuding and well–emphasized secondary phase, abundant in all eight samples. As challenging as it is to detect secondary phases near the main kesterite peak, one can tell that ZnSe revealed itself most abundantly at higher Bragg angles. A potential presence in the immediate domain of dominant CZTSe peaks, was also ruled out as a result of this perceived closeness.

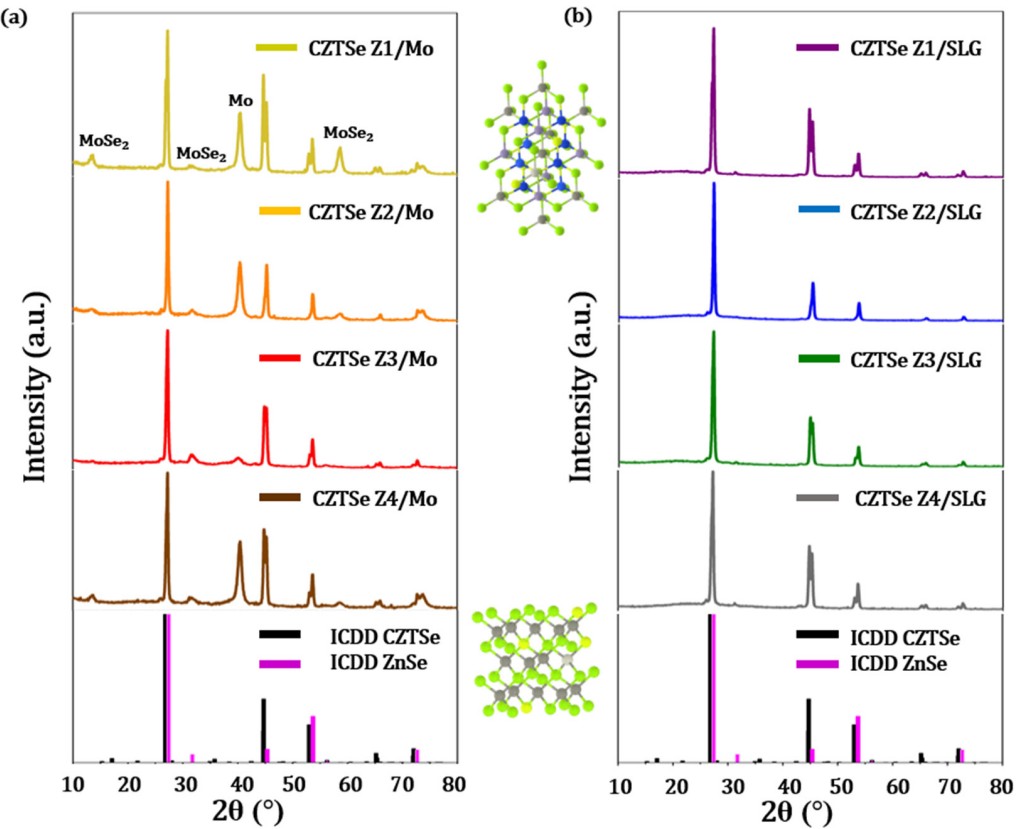

**Figure 2.** Diffractograms plotted using the GIXRD–collected data from the (**a**) Mo–deposited and (**b**) SLG–deposited CZTSe samples alongside ICDD cards associated with the main CZTSe phase and the most prominent secondary phase (ZnSe). The crystal structures of kesterite and ZnSe can be noticed in the middle up and middle down sections with the following color legend: blue dots for Cu(2+), gray for Zn(2+), purple–gray for Sn(2−) and green for Se(2−) atoms.

The peaks observed only in the case of Mo–coated substrate near 13.6°, account for Mo–Se compounds, $MoSe_2$ being widely encountered at back contacts [30]. Other Mo–Se associated peaks (at 31.6° and close to or above 56°) predicted by literature popped up in the diffractograms (Figure 2a). The prominent 40.5° peak clearly signals up an Mo signature, a strong telltale sign of the reflection of the probing radiation on the substrate. As one can observe, the Z3 Mo peak is much flatter and weaker than the signals coming from the other three samples. Thus, the intensity of Mo peaks declines with an increase in the Sn content. This apparent dependence appears owing to the influence of Sn on the reaction between Mo and Se [31]. However, we can straightforwardly and intuitively conclude that, as the Sn concentration increases, the material layer covering the Mo–substrate becomes thicker, thus screening it. Consequently, this translates into weaker Mo–associated signals. Additionally, the formation of a thick $MoSe_2$ layer at the interfacial contact of Mo\CZTSe, though not well understood, could enhance the electrical contact and, as a result, the electrical resistance of a solar cell. Therefore, it is with great interest that the influence of such contact–forming layers on efficiency performance will be studied in the future [30,32].

To gain more insight into the crystallographic structure of the thin films, a crystallite dimensional analysis has been conducted. The size of the crystallites is synthesis temperature dependent, hence the influence of occurring secondary phases on the lattice ordering. The average size of the crystallite in each sample was computed using the Scherrer equation [33]:

$$D = \frac{K\lambda}{\beta\cos\theta} \qquad (1)$$

where K is the shape factor, conventionally taken as 0.9 [34], λ = 1.5406 is the wavelength of the CuKα radiation, β stands for the line broadening at half the maximum intensity (FWHM) determined with a Gauss fit and θ represents the maximum diffraction Bragg angle. To this general approach, there was added an instrumental broadening contribution for a more precise estimation of D. Thus, we used a modified Scherrer equation that beside these broadenings, taking into consideration the corrections associated with the peak fit and the emitted radiation wavelength [35]. Figure 3 provides the average crystallite size histograms. A decreasing trend in the crystallite dimensions, over the entire value span, can be notices, outlining the impact of synthesis conditions on morphology and crystallographic properties. Beginning from 9.79 nm and extending to 18.24 nm in the SLG case and from 12.02 nm to 24.92 nm for Mo substrates, the broad domain of values backs up evidence on the effects of physical parameters during synthesis and the precursor target nature on crystalline ordering. The CZTSe/Z1 (on both Mo and SLG) stack annealed in two different atmospheres, prior to each deposition, exhibits the largest average size among their designated substrate categories. Since such a deposition stacking scheme is supposed to yield Se–deficit samples (as compared with enabled schemes in the other samples), one can empirically conclude that the apparent scarcity of Se influenced the results.

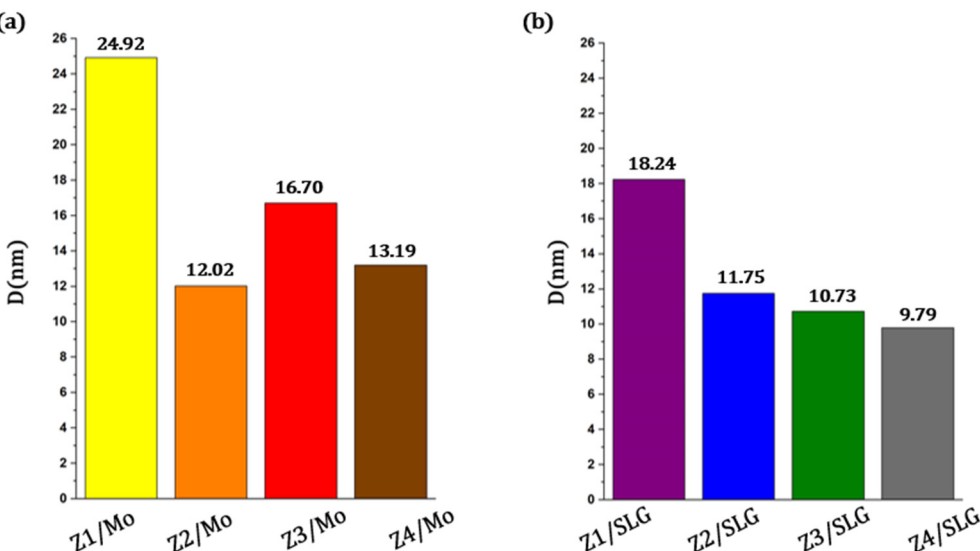

**Figure 3.** Average crystallite size in each of the (**a**) Mo–deposited and (**b**) SLG–deposited CZTSe samples. The color chart remains the same for all graphical representations and histograms in the manuscript, namely: yellow is assigned to CZTSe/Z1/Mo (or, here, Z1/Mo), orange resembles CZTSe/Z2/Mo, red, CZTSe/Z3/Mo and brown, CZTSe/Z4/Mo, whilst purple stands for CZTSe/Z1/SLG, blue for CZTSe/Z2/SLG, green for CZTSe/Z3/SLG and grey for CZTSe/Z4/SLG.

*3.2. Raman Spectrscopy*

Figure 4 displays the Raman spectra of the eight CZTSe samples succeeding the final annealing at 550 °C. The main kesterite CZTSe phase is noticed at the highest peaks corresponding to Raman shifts of 173, 197 cm$^{-1}$, followed by weaker counterparts at 232/238, 240/242 cm$^{-1}$. The last cited peak (240/242 cm$^{-1}$) could also be assigned to MoSe$_2$ [36,37], thus confirming the GIXRD detection. ZnSe traces are observed near 251/254 cm$^{-1}$ (cubic ZnSe [38]). The right–shifted 258/260 cm$^{-1}$ peaks could appear due to an excess of Cu [39] not necessarily due to extra Zn and a broadened spectrum [38,39]. Other sources outline at this shift traces of Cu$_{2-x}$Se [36,37]. Our results cannot conclude this assumption, because Cu–Se compounds were not directly observed in GIXRD. One can also note the asymmetrical profile of the main peaks that broadens towards the lower Raman shifts, asymmetry discerned in all Mo and SLG deposited films. Similar profiles have been observed in a series of CZTSe–based studies [39] that lead to a thread of assumptions and

reported presence of stannite/PMCA–crystal modification [38,40] or disordered kesterite in the considered structure. Moreover, the contribution of the 190/191 cm$^{-1}$ lines can also be linked to the aforementioned broadening effect. Furthermore, these lines could be assigned to CTSe [39]. SnSe or SnSe$_2$ phases have been linked to this line as well in previous studies [40,41] but, in our case, for seemingly Sn–deficit films, this assumption was fully ruled out. However, the large differences from the predicted values observed in these broadened shifts, might be caused by disordered kesterite [38]. Debates have sprung up over the dominant Raman peak broadening effect and future research has to be conducted for more accurate binary and ternary phase identifications.

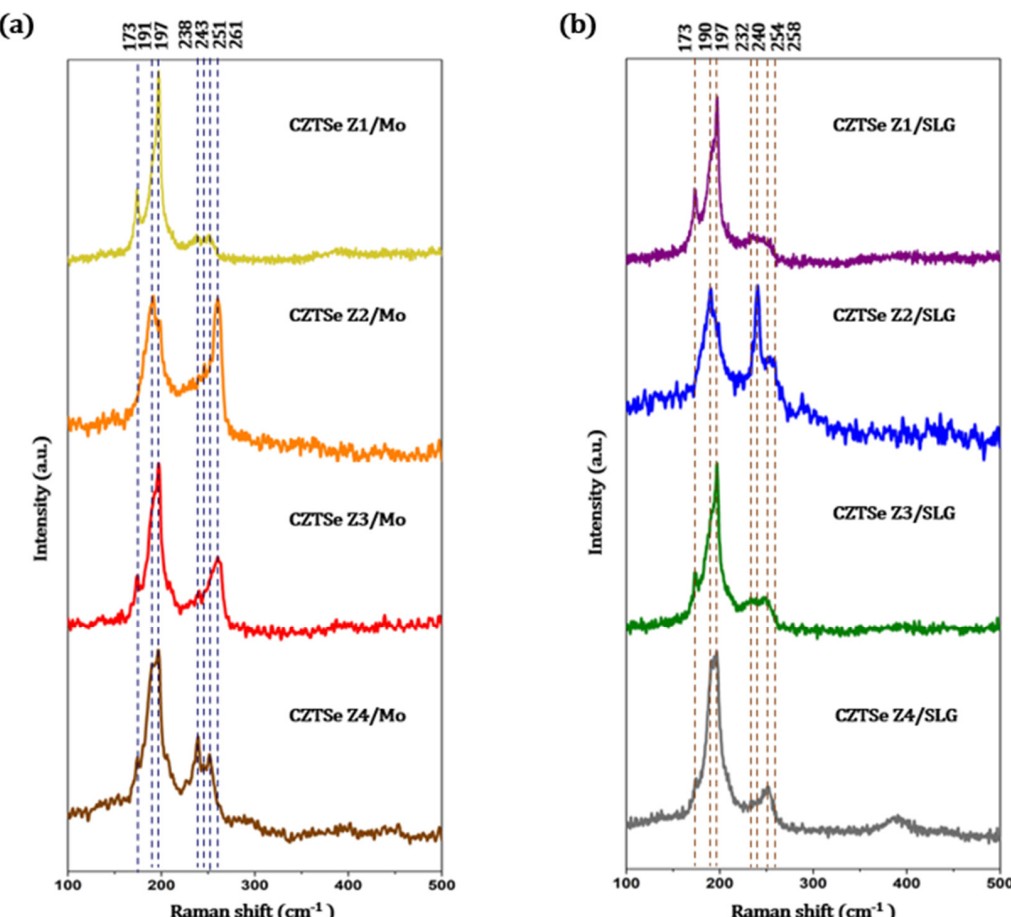

**Figure 4.** Raman spectra and the shifts corresponding to the main and secondary CZTSe phases in the 100–500 cm$^{-1}$ wavenumber region for the (**a**) Mo–deposited and (**b**) SLG–deposited samples.

### 3.3. SEM Images and EDS–Measured Elemental Concentrations

Images of the surface morphology recorded via SEM yields out information about the structure of the films, grain size, faults, cracks, plate–like structures, voids, cavities and other defects. Larger grains are correlated to reduced recombination losses in the production of photogenerated electrons and a widened carrier diffusion length, thus a future solar cell with such embedded films would be prone to achieve a better conversion efficiency [8,42]. However, a heterogenous surface with micro–defects and off–stoichiometry concentrations can disrupt the results. Figure 5 delivers the morphological survey of the CZTSe samples. CZTSe Z1 (on both substrates) exhibits a dense composition, with the average grain diameter around 1 μm and the largest agglomerations grains close to 2.5–3 μm and background smaller formations. The SLG–deposited Z2 is dominated by 3 μm to 3.75 μm long agglomerations of grains, culminating with a protruding one, to the mid left, of ~5 μm and another at the mid–top of 4.5 μm, that have to be a result of

material clumping and coexisting secondary phases. These two agglomerations have an irregular, elongated, clumped–up shape, meaning they originate from secondary phases and compounds fusing together. In the Mo–covered Z2 sample, filament–looking formations, rather sparse in the upper area, are present. In the CZTSe/SLG Z3, we observe an elongated agglomeration of grains about 5 μm long followed by a 4 μm one and the rest of mid–sized and smaller crystalline structures. On the other hand, the Mo–contact film has a packed and homogeneous morphology. Finally, the film with the most regular–sized grains is Z4, on both the SLG and Mo substrates. Several voids can be seen in the images as well. As it has been argued within the XRD and Raman analysis sections, the morphologies are heavily affected by the film growth conditions and subsequent arising secondary phases. Grain boundary defects at the Mo\CZTSe interface are believed to also affect the structural ordering and orientation in the crystal lattice, ultimately influencing the bandgap [43]. Assuming a similar way of thinking to previous studies, our samples (with plenty of ZnSe inclusions) could exhibit such defects.

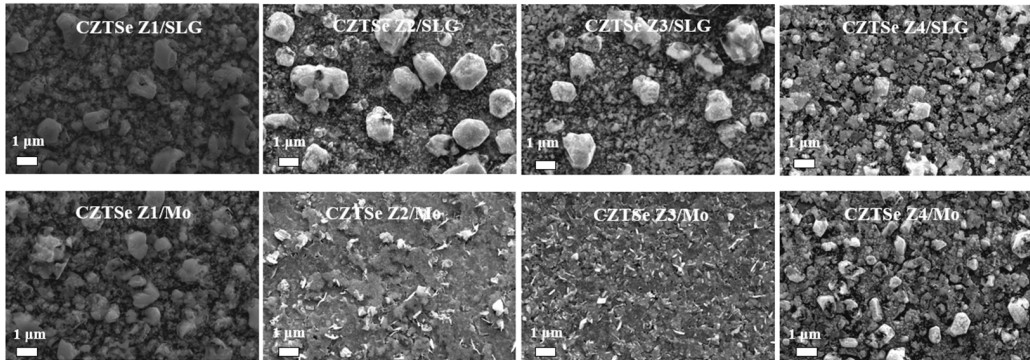

**Figure 5.** SEM images of the CZTSe films.

For a proper examination of the concentration of the films, the eight EDS spectra were plotted and the elemental identification was performed, as can be observed in Figure 6. Supplementary, we introduced an elemental histogram, depicted by Figure 7, to visually confirm these findings and a data table (Table 2) filled with the elemental ratios. The numbers highlight the abundance of Cu. As it has been inferred from the previous physical analysis (GIXRD and Raman), Sn scarcity is due to evaporation during high–temperature post–deposition selenization. Zn is found in larger amounts than stoichiometric ratios. Also, Se has a much smaller contribution (close to 3/5) than its expected phase–pure counterpart to the overall stoichiometry, due to evaporation as well. Therefore, Sn and Se are evaporated as SnSe compounds. ZnSe inclusions were encountered before in our samples, owing to the post–deposition selenization interlayer reactivity.

**Table 2.** Data table providing the ratios of Cu, Zn, Sn and Se after the last 30 min–long post–deposition selenization at 550 °C. These values are calculated as an average over three scanned regions.

| Elemental Concentration (%) | CZTSe/Mo | | | | CZTSe/SLG | | | |
|---|---|---|---|---|---|---|---|---|
| | Cu | Zn | Sn | Se | Cu | Zn | Sn | Se |
| Z1 | 44.72 | 22.46 | 3.09 | 29.73 | 40.39 | 22.26 | 5.96 | 31.39 |
| Z2 | 21.65 | 32.84 | 5.99 | 39.52 | 22.48 | 32.08 | 11.45 | 33.99 |
| Z3 | 36.47 | 22.16 | 6.36 | 35.01 | 31.19 | 26.58 | 10.26 | 31.97 |
| Z4 | 39.81 | 25.79 | 3.75 | 30.65 | 38.14 | 23.95 | 6.94 | 30.97 |

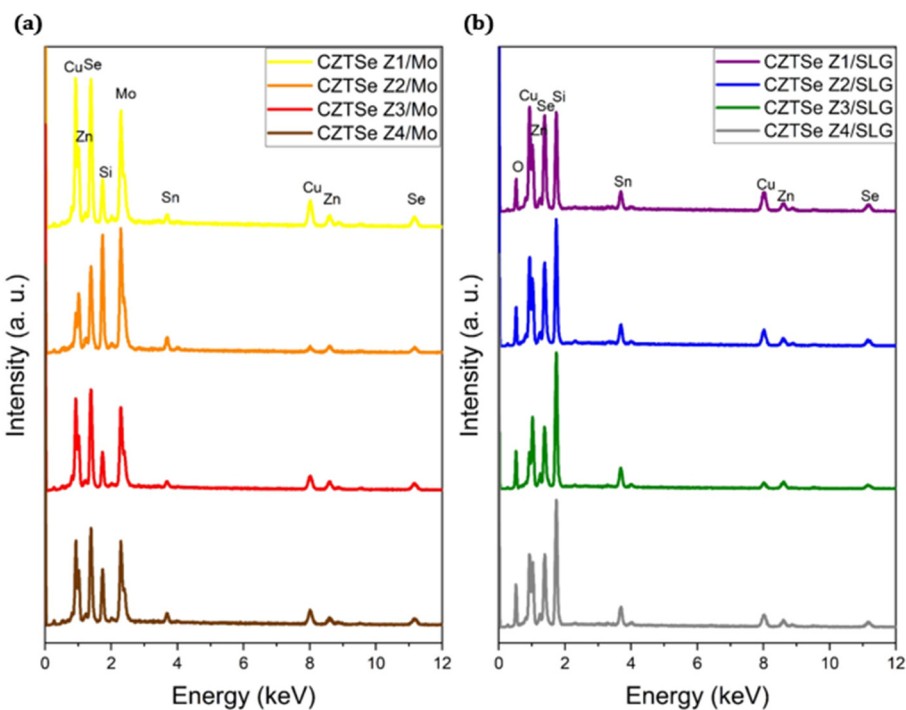

**Figure 6.** EDS spectra and elemental identification for the (**a**) M–coated and (**b**) SLG–coated samples.

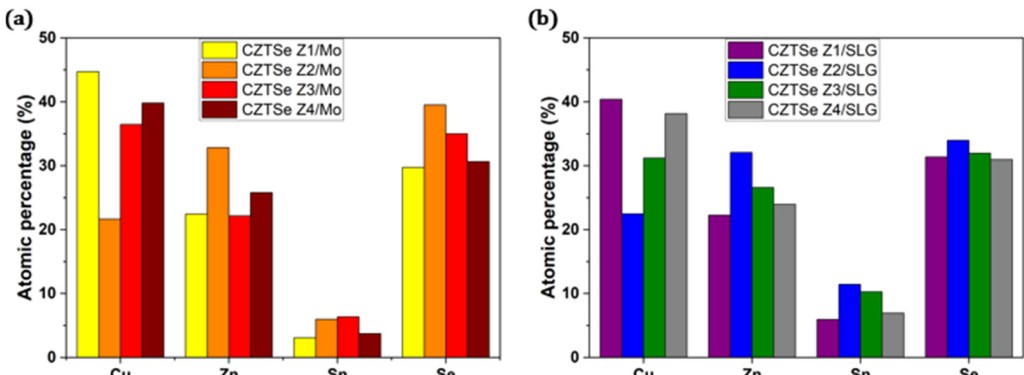

**Figure 7.** Histogram of the elemental ratios as calculated from the EDS measurements for the (**a**) Mo–deposited and (**b**) SLG–deposited samples.

Additionally, our histogram resembles a visual proof of the elemental composition concentration. The off–stoichiometric data points out the high Cu and Zn concentration, an unwanted result for photovoltaic practices. Theoretically, a Cu–poor and Zn–rich structure provides the stabilization of a deep acceptor layer that confers advantageous electronic and conduction properties [44,45]. One can agree that the second annealing procedure had a rather harmful impact on the Sn– and Se–atomic ratios. Low pressure post–deposition selenization impedes the reactivity and cohesion between Sn and the other elements and the high temperature of 550 °C promotes internal CZTSe decomposition and aformentioned Sn volatility [46]. However, as a comparative observation, CZTSe Z3/Mo appears to have a relatively higher content of Sn than the other three Mo–deposited films, in agreement with the assumption stated in the previous section [31].

### 3.4. Optical Measurements and the Determination of the Optical Bandgap

The transmission spectra in the UV–Vis–NIR wavelength range were collected and then employed in the evaluation process of the optical bandgaps of the SLG–deposited samples. After calculating the absorbance coefficient $\alpha$ by using the Beer–Lambert law [47], we

plotted the Tauc graphs, of $(\alpha h \nu)^2$ vs. $h\nu$, and found the resulting bandgap by extrapolating the linearized region of the graph. $\alpha$ varies between 2.45 and 6.92.

The Tauc plot method states that [26]:

$$(\alpha h \nu)^{1/n} = C(h\nu - E_g) \tag{2}$$

wherein $h\nu$ is the energy of the incident photon, the exponent $1/n$ (as CZTSe materials exhibit directly allowed transitions, $n = 1/2$.) describes the transition type inside the semiconductor, C is a numerical constant and $E_g$ represents the bandgap.

Figure 8 displays the plots and values of the bandgaps, namely 1.49 eV for Z1, 2.01 eV for Z2, 1.63 eV for Z3 and 1.42 eV for Z4, on the SLG substrate. The decreased absorbance and, reciprocally, absorption come as a consequence of synthesis conditions and coexisting secondary phases [32]. Thus, the bandgaps suffered a widening effect, exceeding the optimal values predicted by literature.

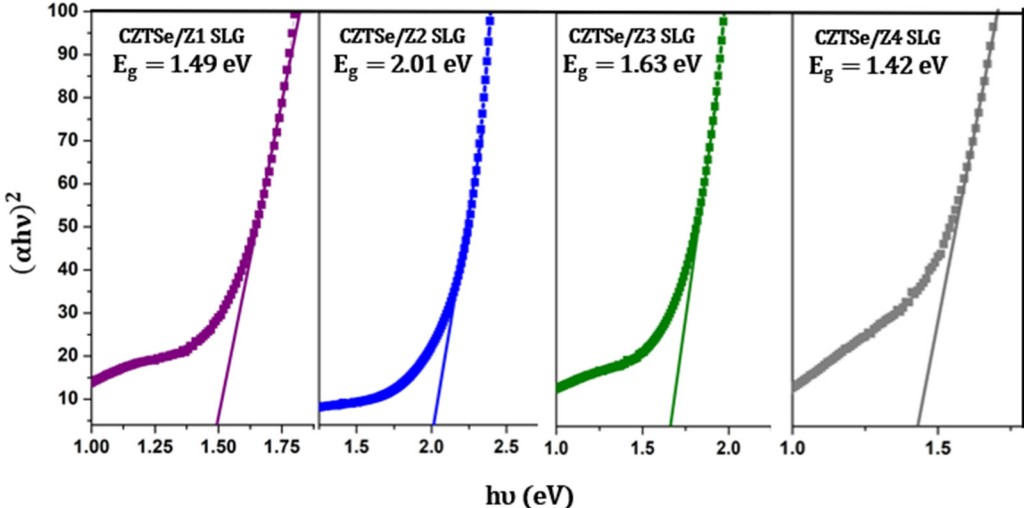

**Figure 8.** Tauc plots used for the calculation of the optical bandgaps of the SLG–deposited CZTSe samples.

A past reported conundrum concerning the optical bandgaps consists of the discrepancies of the transmission–inferred $E_g$ (around 1.5 eV) in comparison with other calculation based on the overall EQE, photoluminescence and theoretical methods that provided the commonly considered value of 1 eV [48–50]. Employed preparation techniques, the choice of experimental setup, the extrapolation or calculation procedure of the optical bandgap and the identification of coexisting secondary phases that might enlarge its value might be at the very root of these results. ZnSe, the most abundant secondary phase in our samples, might bear a significant impact on the widening effect [50], leading to values near 1.5 eV, as two of our samples (Z1 and Z4) already displayed.

However, the accepted view on the matter remains that there is an $E_g$ in the 1 eV range for CZTSe materials, but the presented results show overall higher values, due to the off–stoichiometric nature.

Evidently, two post–deposition heat treatments, with the last annealing process having the greatest impact, established a huge contributor to the energy band characteristics in thin films. As the driving force behind interlayer reactivity, crystalline growth, interface porosity and grain structure, high–temperature post–deposition annealing alters the interatomic structure and poses lattice defects, thus widening the bandgap.

## 4. Discussion

Effectively controlled synthesis steps could mitigate the formation of multiple secondary phases, ensuring denser and more uniform structures with larger crystallites and

interatomic spacing. Achieving a pure $Cu_2ZnSnSe_4$ kesterite phase is a challenging task for most sputtering techniques and both in situ and ex situ post–deposition selenization procedures. In situ post–deposition heat treatments with temperatures ranging between 300 and 550 °C paved the road towards adequate opto–electronic results [23], here mentioning a study attempt [27], carried out at 400 °C yielding out compact films with a bandgap of 1 eV. A solution that might aid the formation of secondary phases would be a continuous thermodynamic parametric control, where pressure alongside temperature evolution inside the furnace can be cautiously tailored. The Se–vapor concentration and reaction rate with the annealed layer are dependent on the Ar flow and furnace–enabled parameters; suppression of the Se vapor–related kinetics during low–temperature regimes might halt binary Cu–Se compounds. Accordingly, a faster temperature ascent towards the steady >500 °C plateau prevents the occurrence of intermediate phases [27]. For Mo–coated substrates, imposing a limit for the thickness span of arising $MoSe_2$ interfacial layers could be enacted by the formation of Cu–Zn and Cu–Sn microstructures. In effect, these binary micro–formations could limit the reactions between Mo and Se throughout a high annealing temperature regime [32,51,52]. Obviously, imposing a limit on the Cu atom population should be attained by upgrading stack–deposition either by hybrid co–deposition [29] SEAL methods [53], or successive depositions similar to our chosen course of action. Diffusion kinetics of Cu atoms and systematic post–deposition selenization should enact uniformity and compactness in kesterite thin films. Hypothetically, an alternative would be to use, from the beginning, only a Cu–poor and Se–rich precursor in a successive sputtering approach, followed by a short annealing (in an Sn+Se–rich atmosphere) with or without doping agents and parameterized structural analysis. Also, diffraction studies with in situ temperature and pressure adjustments could shed a light on crystalline ordering changes and crystallite growth optimization as well. Also, a temperature–dependent GIXRD (or XRD) measurement would help study the phase formation process to control the occurrence of detrimental secondary phases. On the other hand, an enhanced high–quality crystal growth for CZTSe and, in general for chalcogenides, could be aided by switching to chemical transport reactions [54,55]. Solution–based fabrication methods involving one or more solvents is still a novel synthesis procedure but it has been deemed affordable and eco–friendly, if the use of toxic substances, like hydrazine for example, can be avoided [54]. In their study, Muslih et al. proposed a post–fabrication selenization scheme and rendered promising values for the potential power conversion efficiency and recorded short–circuit current density. Moreover, the use of chemical vapor transport methods in chalcogenides, mentioning the close relative of our studied compound, CZTS, set forth relevant insights in the dependance of crystallization kinetics on thermochemical (isothermal or temperature gradient–induced studies) conditions [55].

## 5. Conclusions

This study introduced detailed insights on the mechanisms behind secondary phase formation and observed morphology in CZTSe absorber layers and their deleterious effect on the nature of highly–sought semiconductor properties. High–temperature post–deposition heat treatments in Sn+Se and Se–rich environments stabilized the precursor components. The kesterite phase was detected by the 27.28° main peak in GIXRD, while Raman measurements confirmed the appearance of the phase at 197 $cm^{-1}$ shift. ZnSe signals were charted out from the GIXRD diffractograms and the Raman spectra as well. Presumably, scarce traces of other secondary phases like $Cu_2Se$ and CTSe were present alongside the CZTSe phase. With much higher concentrations of Cu atoms and low Se, our samples exhibit an off–stoichiometric elemental configuration. Sn has a very small contribution for the total concentration, owing to accentuated evaporation during the last high–temperature post–deposition annealing. With only two samples exhibiting desirable optical bandgaps, we drew out a band–gap broadening effect resulting from structural defects and off–stoichiometric compositional data. Future experimentally driven studies

and qualitative research need to be performed in order to make progress in the field of synthesis technologies of pure phase CZTSe films.

**Author Contributions:** Conceptualization, A.V., D.-S.C. and M.Y.Z.; methodology, A.V.; validation, A.V., M.Y.Z., I.-D.S., A.-T.B., F.S. and D.-S.C.; formal analysis, A.V., D.-S.C., M.Y.Z., I.-D.S., A.-T.B. and F.S.; resources, A.V., D.-S.C., A.-T.B., F.S. and I.-D.S.; data curation, M.Y.Z., D.-S.C., A.-T.B., F.S. and I.-D.S.; writing—original draft preparation, D.-S.C., M.Y.Z. and A.V.; writing—review and editing, D.-S.C., M.Y.Z. and A.V.; supervision, A.V.; project administration, M.Y.Z. and A.V.; funding acquisition, M.Y.Z. and A.V. All authors have read and agreed to the published version of the manuscript.

**Funding:** The authors acknowledge the financial support provided by the Romanian Government through the Executive Unit for Financing Higher Education, Research, Development and Innovation (UEFISCDI), under the Romanian Ministry of Research, Innovation, and Digitalization. This support was received within the framework of the following projects: PN–III–P1–1.1–PD–2021–0240 (contract no. PD 41/2022) and Core Program PC3–PN23080303.

**Institutional Review Board Statement:** Not applicable.

**Informed Consent Statement:** Not applicable.

**Data Availability Statement:** The data presented in this study are available in this article.

**Conflicts of Interest:** The authors declare no conflict of interest.

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
