# Peer review of "Understanding the Effects of Post-Deposition Sequential Annealing on the Physical and Chemical Properties of Cu2ZnSnSe4 Thin Films"

_surfaces, doi:10.3390/surfaces6040031_

Round 1
Reviewer 1 Report
Comments and Suggestions for Authors
1. The first appearance of CZTSe requires a full name in the abstract.
2. There are two small images in Figure 2-4 that need to be labeled as a and b.
3. The conclusion section was written too long.
4. I suggest testing the electrical properties of the CZTSe film.
Author Response
Theank you for your comments. Answers are included in the attached file.

Reviewer 2 Report
Comments and Suggestions for Authors
In their contribution Understanding “The Effects of Post-Deposition Sequential Annealing on the Physical and Chemical Properties of CZTSe Thin Films”, Catana et al. report on the results of their explorations of the quaternary systems Cu-Sn-Zn-Se. In the framework of their contribution, the authors describe the syntheses of the inspected samples, which were obtained as thin layers, while different means were employed to determine the compositions and the structural features. In addition to these explorations, the authors also reported on the experimentally determined band gaps. Although the contents presented in that work agree well with the scope of Surfaces, yet, there are certain issues, which should be solved prior to a publication of the contribution:
- As indicated by the authors, the quaternary phases were not obtained as pure phase; however, there is an additional reflection (at around 40°) that is of rather high intensity in the diffraction diagrams of the samples Z1, Z2 and Z4 and has not been assigned to any phase. Do the authors have any idea with regard to the composition of the side-product? How does that side-product influence the other measurements?
- It is stated that the compositions were determined by means of EDS; however, I could not find any spectra. Therefore, I strongly recommend to include that relevant data as Supporting Material. Furthermore, the authors should provide further information regarding the EDS device, because the determinations of both Cu and Zn within a given samples require highly accurate EDS measurements.
- With regard to the description of the crystal structure, it will be helpful if the authors include a representation showing the respective crystal structures.
- In the framework of their explorations, the authors determined the band gaps of the respective samples; however, it is quite hard to say how the different components within the samples influence the widths of the band gaps. As the compositions of the samples are not entirely clear, I suggest to remove the discussion regarding the band gap and include it as part of future publications reporting on the experimental results of pure phase samples.
- It is mentioned that it is quite difficult to obtain pure phase samples. Maybe, an alternative approach could help: it is possible to grow large single crystals by employing chemical transport reactions. Accordingly, it should be possible to obtain large single crystals of the target compound by means of such transport reactions. The single crystals should not contain any impurities and could easily be manipulated in order to obtain thin layers. On the other hand, it should also be possible to manipulate the temperature program of such a transport reaction so that rather thin single crystals are obtained.
- There are also certain typos (e.g. Brucker), which should be corrected. Furthermore, the authors should also mention if special software has been used.
Comments on the Quality of English LanguagePlease check the last point of the attached review.
Author Response
THank you for your comments. Answers are given in the attached file.

Round 2
Reviewer 1 Report
Comments and Suggestions for Authors
The author has been revised the paper. I advise to accept it.
Reviewer 2 Report
Comments and Suggestions for Authors
In the revised version of their contribution, the authors have solved all problems as pointed out by the reviewers. Therefore, I recommend to publish the contribution as it is; yet, there are still certain open questions which can be answered as part fo future research.